# Gold-Polypyrrole-Loaded Eosin in Photo-Mediated Treatment of Hidradenitis Suppurativa: In Vivo Trans-Epidermal Permeation Study and Clinical Case Report

**DOI:** 10.3390/pharmaceutics14102197

**Published:** 2022-10-15

**Authors:** Abdullah I. El-Kholy, Maha Fadel, Maha Nasr, Ibrahim El-Sherbiny, Abeer Tawfik, Yasser O. Mosaad, Doaa Abdel Fadeel

**Affiliations:** 1Department of Medical Applications of Laser, Pharmaceutical Nano-Technology Unit, National Institute of Laser Enhanced Sciences (NILES), Cairo University, Giza 12613, Egypt; 2Department of Pharmaceutics and Industrial Pharmacy, Faculty of Pharmacy, Ain Shams University, Cairo 11566, Egypt; 3Nanomaterials Lab, Center for Materials Science, Zewail City of Science and Technology, 6th October City, Giza 12578, Egypt; 4Department of Medical Applications of Laser, Dermatology Unit, National Institute of Laser Enhanced Sciences (NILES), Cairo University, Giza 12613, Egypt; 5Department of Pharmacy Practice and Clinical Pharmacy, Faculty of Pharmaceutical Sciences and Pharmaceutical Industries, Future University, Cairo 11835, Egypt

**Keywords:** polypyrrole, gold, eosin yellow, photodynamic, photothermal, Hidradenitis Suppurativa

## Abstract

This study reports a new protocol for the management of Hidradenitis Suppurativa (HS), depending on the synergistic photodynamic and photothermal effect of eosin yellow-gold-polypyrrole hybrid nanoparticles (E-G-Ppy NPs). E-G-Ppy NPs and gold-polypyrrole NPs (G-Ppy NPs) were synthesized, characterized, and formulated in topical hydrogels. Then, in vivo trans-epidermal permeation study, under both dark and white light-irradiation conditions, was done on albino mice. The E-G-Ppy hydrogel was then applied on a twenty-four years old female with recurrent axillary HS lesions pretreated with fractional CO_2_ laser. Thereafter, the treated lesions were irradiated sequentially, using an IPL system, in the visible (~550 nm) and NIR band (630–1100 nm) to activate the synthesized nanoparticles. Results showed that, upon application to mice skin, E-G-Ppy exhibited good tolerance and safety under dark conditions and induced degenerative changes into dermal layers after white-light activation, reflecting deep penetration. Photo-activation of E-G-Ppy hydrogel to a severe Hidradenitis Suppurativa case showed an improvement of 80% of the lesions according to average HS-LASI scores after 4 sessions with no recurrence during a follow-up period of six months. In summary, the dual photodynamic/photothermal activation of E-G-Ppy NPs can represent a promising modality for management of HS. Further expanded clinical studies may be needed.

## 1. Introduction

Hidradenitis Suppurativa (HS), is an inflammatory recurrent dermatosis affecting different areas of the skin. It is characterized by the presence of thick scarring, oozing abscesses, and painful nodules in different areas such as armpits, buttocks, groin, or below the breast [1,2]. Causes of HS are not well established, but potentially thought to be mainly related to apocrine glands problems. Yet, HS may also be related to other factors such as hair follicle structural anomalies, obesity, and smoking, among others [3]. HS is commonly treated by oral/topical antibiotics, oral/topical isotretinoin, anti-inflammatory drugs, or surgery [4]. Photodynamic therapy (PDT) was studied as a promising new technique for the treatment of HS; with encouraging results obtained [5,6]. PDT depends on the activation of a photosensitive compound (a photosensitizer; PS) by light of an appropriate wavelength in the presence of oxygen. The activated PS returned to its ground state liberating free radicals and/or reactive oxygen species that cause tissue destruction [7,8]. Previous studies reported the use of PSs as 5-aminolevulinic acid (ALA) [2], and methylene blue [4,6] for PDT treatment of HS. The main drawbacks were the low penetration of the PS, which can be overcome by the proper choice of a delivery system [4]. In a previous work, highly dispersible, ultra-nanonized, and highly stable polypyrrole-coated gold nanoparticles (G-Ppy NPs) were synthesized and showed promising photothermal effects. In that method of synthesis, chloroauric acid was used as an oxidizing agent to induce the oxidative polymerization of pyrrole into polypyrrole, resulting in the formation of gold-conjugated polypyrrole with highly enhanced pharmaceutical properties in comparison to polypyrrole synthesized via the conventional methods [9]. Later, we conjugated two hydrophilic PSs; eosin yellow (E) and rose bengal (RB) with the synthesized nanoparticles and studied their combined photothermal and photodynamic effect in vitro using HeG2 as a model for cancerous cell lines [10]. Eosin yellow is a water-soluble dye that we used successfully as a photosensitizer in topical PDT of recalcitrant plantar warts [11] and onychomycosis [12]. The goal of this work is to report our experience with a new protocol in a case report for HS management that depends on the combined synergistic photodynamic and photothermal effect of the synthesized (E-G-Ppy NPs). The synthesized nanoparticles were formulated as a topical hydrogel and evaluated in vivo for trans-epidermal permeation (on albino mice skin) before application on a patient with HS (clinical case report).

## 2. Materials and Methods

### 2.1. Materials

Eosin yellow sodium salt, chloroauric acid (HAuCl_4_), and pyrrole were purchased from Sigma-Aldrich Corp. (St. Louis, MO, USA). Sodium-carboxymethylcellulose (Na-CMC) was purchased from Normest Company for scientific development, Egypt. All other reagents and solvents were of analytical grade.

### 2.2. Synthesis of Free and Eosin-Conjugated Gold-Polypyrrole Nanoparticles (G-Ppy and E-G-Ppy)

Gold-polypyrrole nanoparticles loaded with eosin were synthesized as described in a previous paper [10]. Briefly, aqueous pyrrole solution (25 mM) was mixed with an aqueous eosin solution (0.5 mM). A solution of 1 mL HAuCl_4_ (10 mM) was added to 20 mL of the pre-prepared pyrrole-eosin solution, under stirring for 4–5 min, followed by centrifugation at 10,000 rpm to separate any byproducts and unreacted pyrrole. The precipitated nanoparticles were re-dispersed in 20 mL double distilled water and left as a stock colloidal solution. G-Ppy NPs (free of E) were also synthesized in the same molar ratio (HAuCl_4_: Pyrrole at MR: 1:50, respectively) for comparison, according to the procedure previously described by Fadel et al. [9].

### 2.3. Characterization of G-Ppy NPs and E-G-Ppy NPs

Morphological characteristics of the synthesized G-Ppy NPs and E-G-Ppy NPs were examined by transmission electron microscope TEM (JEOL Ltd., Tokyo, Japan). Zetasizer (Malvern Ltd., Malvern, UK) was used for particle size and zeta potential measurements. A double-beam spectrophotometer (Rayleigh UV-2601, Beijing, China) was used to study the UV–visible-NIR absorption spectra of eosin yellow and the synthesized nanoparticles.

### 2.4. Topical Hydrogel Preparations of E, G-Ppy NPs, and E-G-Ppy NPs

Free E, free G-Ppy NPs, and E-G-Ppy NPs were all formulated in a 3% carboxymethylcellulose (CMC) hydrogel base as described by Fadel et al. [13]. The concentration of E in free E hydrogel and E-G-Ppy hydrogel was 0.03455% w/w and the concentrations of G-Ppy in G-Ppy hydrogel and E-G-Ppy hydrogel were equivalent. The produced hydrogels were visually examined for appearance and uniformity. The pH values of the produced hydrogels were determined using a digital pH meter (HANNA, RI, USA). Samples from different areas of the free E and E-G-Ppy NPs hydrogels were taken, diluted with distilled water, and measured colorimetrically at 515 nm for the study of hydrogel uniformity and the actual E concentration.

### 2.5. In Vivo Evaluation of Trans-Epidermal Permeation Following the Application of G-Ppy NPs and E-G-Ppy NP Hydrogels

Trans-epidermal permeation was studied by tracking the histological changes in skin layers after the application of hydrogels in dark and after irradiation of the animal’s skin. This experiment was carried out following the same protocol as described by Fadel et al. [13] on male *Mus musculus* albino mice, applying the regulations of Cairo University Institutional Animal Care and Use Committee (CUIACUC); with the approval number (CU-I-F-83-20). All of the animals’ dorsal skin hair was shaved, and the animals were divided into different groups and treated as listed in Table 1.

Animal behavior was observed during the irradiation time, and animals that suffered from distress signs or pain were administered isoflurane inhalation of low doses. After the experiment, each animal was kept in a separate clean plastic cage for 24 h. Afterwards, the animals were anesthetized and executed. For histopathological examination, the treated skin areas were separated and fixed in 10% neutral buffered formalin for 72 h. They were then processed and 4μm thick tissue skin sections were stained with hematoxylin and eosin and examined by a light microscope (Leica Microsystems GmbH, Wetzlar, Germany).

### 2.6. Clinical Case Study

This case study was carried out at the National Institute of Laser Enhanced Sciences (NILES) clinic, Cairo University, Egypt. Approval number of the board of the research ethics committee of the Institute obtained is (CU-NILES/25/21). The study protocol conformed to the guidelines of the Declaration of Helsinki. We affirm that the patient was provided informed consent for publication of images illustrating the clinical improvement. The study was conducted on a twenty-four-year-old female who presented with recurrent lesions of HS at the axilla on one side for 2 years. The recurrence of the lesions appeared after oral medicines such as isotretinoin and topical clarithromycin. Dermatological examination showed hemispheric nodules with a diameter of 0.3–2 cm that were distributed at and around her axilla. Some of the abscesses became ulcerated with noticed local haphalgesia. The patient’s main complaint was recurrent infection, oozing, and a disfiguring scar. The lesions were of grade 4 according to Hurley classification.

### 2.7. Treatment Protocol

Pretreatment with Fractional CO_2_ Laser before hydrogel application was applied to increase trans-epidermal delivery of E-G-Ppy NPs through the patient’s thick skin. The lesions were subjected to one pass of fractional CO_2_ Laser (Bison, Italy). Fractional CO_2_ Laser parameters were as follows; scanning area 3 ~ 20 mm, fluence: 7.5 mJ/cm^2^, pulse duration: 500 µsec, and dot density: 0.8. Then, 1 g ± 0.2 of E-G-Ppy NPs hydrogel was spread on the lesions as a thin film and applied for 15 min. The lesions were then irradiated, sequentially, using an IPL system (EPI-C PLUS; Espansione Group, Bologna, Italy) using a 550 nm filter and 630–1100 nm wide-band filter, respectively, with 2.5 × 4.5 cm spot area, 20 ms pulse duration, and fluence of 25 J/cm^2^. Sessions were carried out every 2 weeks over a period of 8 weeks. The elimination of the lesions was the treatment endpoint. Photographs were taken before treatment, as a baseline, and after treatment. Treatment efficacy assessment was recorded using the modified Hidradenitis Suppurativa Lesion, Area and Severity Index (HS-LASI) [14] by three independent blinded dermatologists to the intervention protocol. A follow-up period of 6 months was carried out to report any recurrent lesions or side effects.

## 3. Results

### 3.1. Preparation and Characterization of G-Ppy NPs and E-G-Ppy NPs

It was clear from the UV–vis-NIR spectra that conjugation of E to G-Ppy has changed the spectroscopic properties of the E-G-Ppy NPs with an increase in the absorption peak and the appearance of new peaks in the NIR region (Figure 1). TEM imaging revealed the morphologies of both types of nanoparticles which resemble those pre-reported by El-Kholy et al. [10] (Figure 2). The zeta potential of G-Ppy was 45 ± 5.9 mV with an average diameter of 55 ± 6.4 nm and the zeta potential of E-G-Ppy was −33 mV with an average diameter of 33 ± 5.63 nm. Both zeta potential values revealed a good colloidal stability [15].

The prepared hydrogels were clear in color and homogeneous. Hydrogels’ pH values ranged from 6.7 to 6.9 ± 0.5, which are close to skin pH. Therefore, no irritation was expected upon use [16]. The different examined samples were found to retain around 95.3–97.7% of the initially incorporated E, showing acceptable homogeneity and uniformity.

### 3.2. In Vivo Evaluation of Trans-Epidermal Permeation Following the Application of E, G-Ppy NPs and E-G-Ppy NP Hydrogels

The histopathological changes in the skin samples of the studied groups are shown in Figure 3 and summarized in Table 2.

### 3.3. Clinical Outcome

The E-G-Ppy hydrogel was used for the photo-induced treatment of HS. As stated by the patient, there was a marked improvement as the oozing stopped after 2 sessions and the scars were improved. According to average HS-LASI scores reported by blinded dermatologists, an improvement of 80% of the lesion was achieved (Figure 4) after 4 sessions. The patient did not report pain, erythema, or hyperpigmentation during treatment. No side effects or recurrence of lesions were reported during the 6 months of the follow-up period.

## 4. Discussion

In a previous work, we synthesized polypyrrole-coated gold nanoparticles (G-Ppy NPs) by oxidative polymerization of pyrrole mediated by gold ions [9]. The use of polypyrrole as a delivery system has many advantages as it is a conductive biodegradable polymer that can absorb light in the near-infrared (NIR) region, thus producing a photothermal effect [17]. The produced gold-coated polypyrrole differs from conventionally-synthesized polypyrrole in being highly water dispersible which, in turn, makes it suitable for use in pharmaceutical formulations. Besides, the photothermal activity of the synthesized G-Ppy NPs was found to surpass that of polypyrrole synthesized by the chemical conventional method, under the same conditions [9]. Later, we conjugated eosin to the prepared nanoparticles in a one-step method giving eosin-loaded gold-polypyrrole nanoparticles (E-G-Ppy NPs) with high loading efficiency, higher photocytotoxicity, and lower dark cytotoxicity (in comparison to those of either free E or G-Ppy NPs alone); using the HepG2 cell line as a model for cancer cells [10]. In the current study, we extended the work to go deeply into the applications of the synthesized E-G-Ppy NPs. So, we formulated the synthesized E-G-Ppy NPs, as well as free E and G-Ppy NPs, in the form of topical hydrogels and studied their trans-epidermal permeation through skin layers by in vivo investigation of their photo-induced effects on mice skin. Then, we used E-G-Ppy clinically as a dual photodynamic/photothermal treatment in the management of HS. Results revealed that E loaded within G-Ppy penetrated to deep layers and exhibited a pronounced photo-destructive effect that extended to the dermis and the subcutaneous regions if compared to free E and free G-Ppy NPs (Figure 2 and Figure 3, respectively). This can be attributed to the synergism between the photodynamic effect of E and the photothermal effect of G-Ppy NPs, because IPL was applied using two wavelength bands; the first in the visible light (~550 nm; activates E) and the second in a wide band of the NIR region (630–1100 nm; activates polypyrrole). As revealed from UV–Visible-NIR spectra, conjugation of E to G-Ppy NPs led to an increase in the absorption in the NIR region, thus enhancing the G-Ppy NPs’ photothermal effect. Additionally, polypyrrole allows more efficient delivery of eosin into deeper layers due to its polymeric nature, giving more efficiency. In fact, IPL has been used previously to activate 5-aminolevulinic acid (ALA) [18] and methylene blue [4] for PDT treatment of HS. However, in our study, the incubation time was shorter (the hydrogel was applied for only 15 min) which is more convenient for the patient and the improvement was pronounced after four sessions only. The scars of HS were characterized by thick skin. That is why the fractional CO_2_ laser was used prior to the PDT session to assist the trans-epidermal E delivery as it facilitates the penetration and distribution of the hydrogel [18]. Fractional CO_2_ causes ablation of the epidermal layer. The ablative holes may extend to deeper layers generating micro-channels through which the topically applied formula can penetrate and accumulate in deep skin layers [19].

## 5. Conclusions

Eosin-gold-polypyrrole hybrid nanoparticles exhibited good tolerance and safety under dark conditions. Gold-polypyrrole, as a delivery system, proved high efficacy in loading capacity, skin-penetration, and photothermal effect. The latter was concluded from the enhanced photo-mediated effect of gold-polypyrrole-loaded eosin, compared to its free form. Light-activation of topically applied eosin-gold-polypyrrole induced observed damages and degenerative changes through skin layers reflecting deep penetration, as tested on albino mice skin. Clinical application of eosin-gold-polypyrrole hydrogel, after pretreatment with fractional CO_2_ laser, resulted in accelerated improvement of the lesions, proving its combined photodynamic/photothermal effects. In summary, the dual photo-activation of eosin-gold-polypyrrole nanoparticles, after pretreatment with fractional CO_2_ laser, can represent a promising modality for the management of Hidradenitis Suppurativa.

## Figures and Tables

**Figure 1 pharmaceutics-14-02197-f001:**
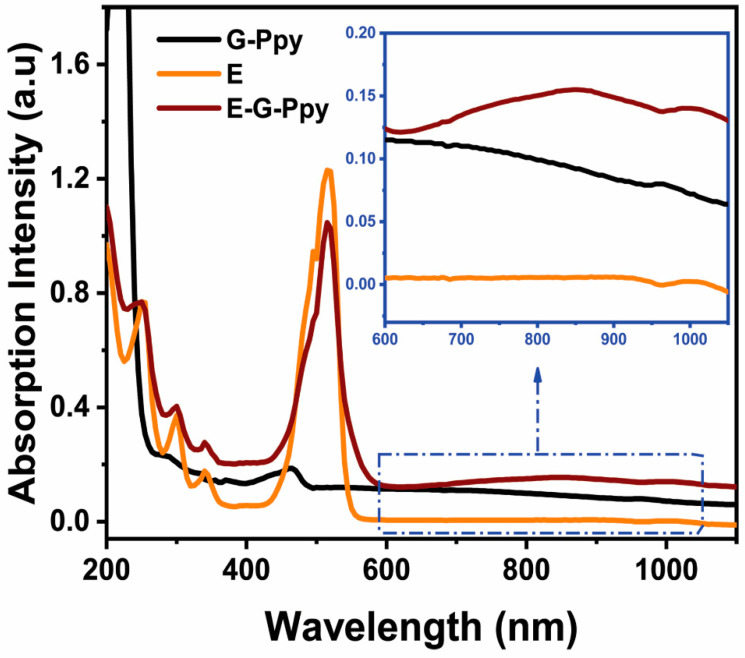
UV–vis-NIR spectra of E, G-Ppy, and E-G-Ppy.

**Figure 2 pharmaceutics-14-02197-f002:**
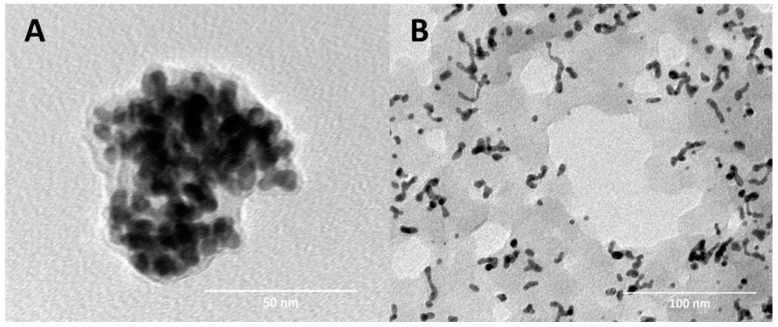
TEM imaging of (**A**) free G-Ppy and (**B**) E-G-Ppy. Preparation and characterization of E, G-Ppy NPs, and E-G-Ppy NPs hydrogels.

**Figure 3 pharmaceutics-14-02197-f003:**
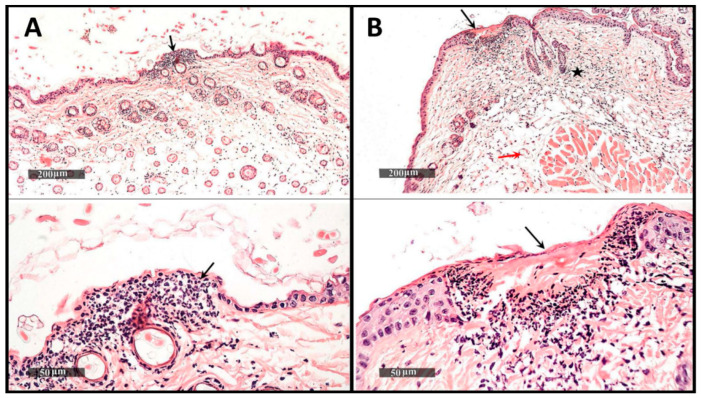
(**A**) Transverse sections of the dorsal mice skin stained (H&E) taken from group 2 (Free E under irradiation): showing the epidermal degenerative changes (black arrows) and (**B**) Transverse sections of the dorsal mice skin stained (H&E) taken from group 3 (E-G-Ppy under irradiation) showing the epidermal degenerative changes (black arrows), ulceration and inflammatory cells in dermal regions (black stars), and the congestive blood vessels (red arrows).

**Figure 4 pharmaceutics-14-02197-f004:**
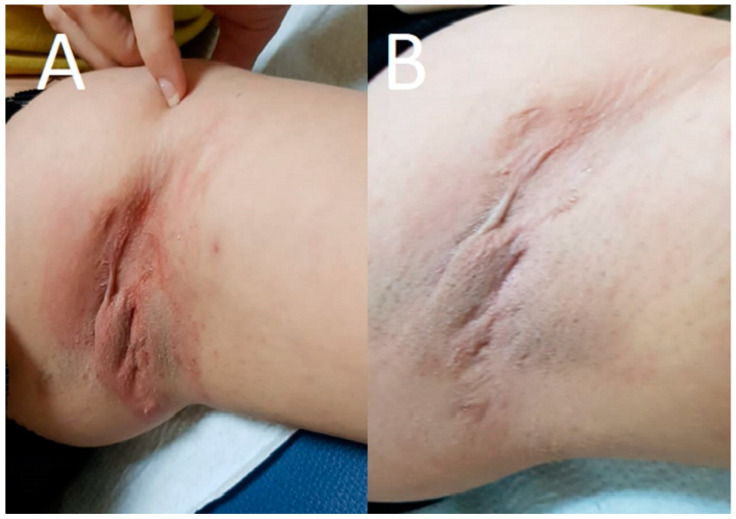
HS lesion (**A**) before treatment showed inflamed oozing hemispheric nodules, and (**B**) after treatment showed improvement to nodules with disappearance of oozing and inflammation.

**Table 1 pharmaceutics-14-02197-t001:** Topical treatment and light irradiation of different animal groups for trans-epidermal permeation study.

Groups (2 Animals per Group)	Topical Treatment *	Light Condition **
Control	**-**	-
Group 1	G-Ppy hydrogel	Dark
		Light
Group 2	Free E hydrogel	Dark
		Light
Group 3	E-G-Ppy hydrogel	Dark
		Light

NB: * Gel was applied for 15 min to a shaved dorsal skin (1 cm^2^ area). ** Animals irradiation was carried out by white light source (Photon scientific, Cairo, Egypt) for 20 min (fluence of 90 mW/cm^2^).

**Table 2 pharmaceutics-14-02197-t002:** Histopathological changes due to trans-epidermal permeation before and after light irradiation of different animal groups.

Group	Epidermal Degenerative Changes	Dermal Ulceration	Inflammatory Cells Infiltrate	Congested Subcutaneous Blood Vessels
Control	-	-	-	-
Group 1-Dark (G-Ppy)	-	-	-	-
Group 1(G-Ppy)-Light	-	-	-	-
Group 2-Dark (free E)	+	-	+	-
Group 2-Light (Free E)	++	-	++	-
Group 3-Dark (E-G-Ppy)	+	-	+	+
Group 3-Light (E-G-Ppy)	+	++	+++	+

Coding system: (-): nil, (+): mild (less than 25% of examined sections), (++): moderate (26–50% of examined sections). (+++): severe (>50% of examined sections).

## Data Availability

The datasets generated during and/or analyzed during the current study are available from the corresponding author on reasonable request.

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
