# Peer review of "Gold-Polypyrrole-Loaded Eosin in Photo-Mediated Treatment of Hidradenitis Suppurativa: In Vivo Trans-Epidermal Permeation Study and Clinical Case Report"

_pharmaceutics, 2022, doi:10.3390/pharmaceutics14102197_

Round 1

Reviewer 1 Report

This report relates to the treatment of dermatitis with photodynamic therapy. It might be interesting for some readers to provide an indication of what can cause this condition and what might be the target(s) for PDT. The photosensitizing agent shows absorbance in the vicinity of 500 nm which means that any effect of light will be limited to a very short penetration into tissues. This may be adequate for the indication.   

It is not made clear what the animal studies revealed. Use of a white light source makes it impossible to calculate the pertinent light dose which will relate only to irradiation in the vicinity of 500 nm. The total light dose was 108 J/sq cm but only a small fraction of this would be pertinent. From the protocol, it appears that normal mouse skin was treated and that there were some effects that are not discussed, aside from a summary in Table 2. What does this mean and how is it pertinent for use of this protocol in a clinical setting?

One case study is reported. This involved preliminary skin abrasion with a CO2 laser, followed by irradiation with light. The description of the light is confusing (lines 131-132). What was the bandwidth of the 550 nm filter? Why 550 nm? It appears that the absorbance of the photosensitizer is centered at 500 nm. What was the purpose of the 630-1100 nm filter?  I assume that both filters were not used at the same time since the 630-1100 nm filter will not transmit 500 nm light. If the filters were used sequentially, this needs to be indicated. This appears to involve the use of a pulsed laser which is otherwise not defined. 

Because of the protocol, it is difficult to decide what did what. Prior use of the CO2 laser could also have had an effect on the outcome. We do not know what this treatment alone might have accomplished or whether irradiation at both wavelengths was needed. So there are no controls and no clear indication of how much of the protocol was needed to produce an effect.  This is clearly a very preliminary study although the overall effect is encouraging. 

The abstract contains a collection of statements that are not fully justified. A protocol was established involving the treatment of a patient with a CO2 laser followed by what appears to be a photodynamic process involving two different wavelengths (although this is not made clear). The net result was a considerable improvement of a pathologic condition although the authors do not know [1] whether the CO2 laser treatment was needed or what this might have accomplished, [2] whether treatment at both 550 nm and in the NIR were needed. The authors need to discuss these points in both the Abstract and the report. 

Author Response

We thank the reviewer for the positive opinion of the manuscript.

-Comment 1:

This report relates to the treatment of dermatitis with photodynamic therapy. It might be interesting for some readers to provide an indication of what can cause this condition and what might be the target(s) for PDT. The photosensitizing agent shows absorbance in the vicinity of 500 nm which means that any effect of light will be limited to a very short penetration into tissues. This may be adequate for the indication.   

-Reply:

We thank the reviewer for this comment. The necessary modifications were added to clarify this point. (Lines: 71-74) “Causes of HS are not well-established, but potentially thought to be mainly related to apocrine glands problems. Yet, HS may also be related to other factors such as hair follicle structural anomalies, obesity and smoking, among others

-Comment 2:

It is not made clear what the animal studies revealed. Use of a white light source makes it impossible to calculate the pertinent light dose which will relate only to irradiation in the vicinity of 500 nm. The total light dose was 108 J/sq cm but only a small fraction of this would be pertinent. From the protocol, it appears that normal mouse skin was treated and that there were some effects that are not discussed, aside from a summary in Table 2. What does this mean and how is it pertinent for use of this protocol in a clinical setting?

-Reply:

We thank the reviewer for this comment. The animal study was mainly conducted to evaluate the degree of dark safety of all formulations (the intrinsic toxicity of each formulation without irradiation) and the efficiency of penetration of formulated nanoparticles before applying to the clinical case. Irradiation was applied empirically, with relatively high dose of white light, to give an indication about the possible light-induced effects of the designed composite (E-G-Ppy). Since the composite to be tested has two main absorption bands; one at ~500-600 nm (correspondent to Eosin yellow and the embedded gold nanospheres) and the other at the NIR region at ~800-1100 (correspondent to polypyrrole), white light was used to cover both bands and to examine the effects of light in the visible and NIR bands. For that, the irradiation dose used in the animal study was much higher than that used in the clinical case study. Concerning table. 2, the complete set of results is available with the corresponding histopathological micrographs, but we preferred not to enclose them to avoid verbiage; since the already embedded micrographs, besides the mentioned table, clearly illustrate the abnormal changes that are more or less similar to those of the un-shown micrographs. However, the micrographs are present and can be added if the matter necessitates.

-Comment 3:

One case study is reported. This involved preliminary skin abrasion with a CO2 laser, followed by irradiation with light. The description of the light is confusing (lines 131-132). What was the bandwidth of the 550 nm filter? Why 550 nm? It appears that the absorbance of the photosensitizer is centered at 500 nm. What was the purpose of the 630-1100 nm filter?  I assume that both filters were not used at the same time since the 630-1100 nm filter will not transmit 500 nm light. If the filters were used sequentially, this needs to be indicated. This appears to involve the use of a pulsed laser which is otherwise not defined. 

-Reply:

We thank the reviewer for this comment. Indeed, two filters of IPL were used sequentially; each filter transmits a light band composed of bundle of wavelengths around the main wavelength. The first filter main wavelength is a 550 nm filter to cover both eosin yellow (~515 nm) and gold nanospheres (~550), while the second one covers the NIR region to benefit from the photothermal effect of polypyrrole. This was rewritten in a clear way in the manuscript (Lines: 177-181). Specifications of the used pulsed laser were also re-defined in a clear way (Line: 178).

-Comment 4:

Because of the protocol, it is difficult to decide what did what. Prior use of the CO2 laser could also have had an effect on the outcome. We do not know what this treatment alone might have accomplished or whether irradiation at both wavelengths was needed. So there are no controls and no clear indication of how much of the protocol was needed to produce an effect.  This is clearly a very preliminary study although the overall effect is encouraging. 

-Reply:

We thank the reviewer for this comment. In fact, CO2 laser step is just to enhance the penetration of the applied gel. However, applying CO2 laser alone without treatment had insignificant effects. We decided to list this step in the protocol to guarantee achieving maximum benefit for the patient according to “Helsinki-ethical principles for medical research involving human subjects”. Thus, this step, alone, is not significant. As mentioned before, the two applied wavelengths were not chosen randomly, but based on the absorption bands of the tested composite. Here, it is useful to mention that the UV-Visible-NIR spectra graph may be a source of misunderstanding. This is because the composite colloidal solution was to be, obligately, diluted to a suitable extent that can be rightly read by the used spectrophotometer (i.e., avoiding overflow). This is due to the high content of Eosin Yellow in E-G-Ppy composite colloidal solution which has intensive absorption, in the visible region, greatly exceeds that of G-Ppy in the Visible-NIR region. Taking in consideration that the concentrations of both components in the composite, upon dilution, are decreased in the same ratio; the absorption of G-Ppy partition seems to be neglectable although the matter is not so. Indeed, it is the high absorption of Eosin Yellow that gives such false impression about the absorption of G-Ppy which surpass that of conventionally-prepared Ppy by more than the triple.1 This may be also the reason for the misunderstanding concerning the use of NIR irradiation. This point was clarified by adding a sub-graph (inset), illustrating the NIR-absorption of the composite, to the original graph (The modified fig. 1, Line: 196-198).

-Comment 5:

The abstract contains a collection of statements that are not fully justified. A protocol was established involving the treatment of a patient with a CO2 laser followed by what appears to be a photodynamic process involving two different wavelengths (although this is not made clear). The net result was a considerable improvement of a pathologic condition although the authors do not know [1] whether the CO2 laser treatment was needed or what this might have accomplished, [2] whether treatment at both 550 nm and in the NIR were needed. The authors need to discuss these points in both the Abstract and the report. 

-Reply:

Thanks for the valuable comment. These points were re-discussed and cleared in the abstract and report (Lines: 44-55) and (lines: 275-280). Protocol issues were already cleared in the previous comment’s reply.

significant effects.

References

1          Fadel, M., Fadeel, D. A., Ibrahim, M., Hathout, R. M. & El-Kholy, A. I. One-Step Synthesis of Polypyrrole-Coated Gold Nanoparticles for Use as a Photothermally Active Nano-System. Int J Nanomedicine 15, 2605-2615, doi:10.2147/IJN.S250042 (2020).

Reviewer 2 Report

The authors synthesized Eosin yellow-gold-polypyrrole hybrid nanoparticles (E-G-Ppy) and evaluated their physicochemical properties, skin permeability, and usefulness after the application of the gel formulation to patients. This study is interesting because it demonstrates the efficacy of the combined photodynamic/photothermal effect of polypyrrole-loaded eosin. The paper also accurately compared the results with previous studies. However, the following comments need to be addressed.

1. With Eosin, the zeta potential changed from positive to negative. Drug penetration into the skin may be highly dependent on the charge of the formulation. The skin penetration and clinical results of the present study should be discussed from the zeta of the formulation.

2. In the description of Figure 3, black stars and red arrows are absent and should be deleted.

3. Figure 4 shows changes in skin properties such as inflammation, but are these results transient? If these changes persist, it is a negative clinical finding. The authors need to explain their findings.

4. In addition, HAuCl4 in the gel formulation is expected to remain in the body as gold nanoparticles after application to the patient. The safety of this formulation must also be demonstrated.

5. The description of the Figure in the Clinical case study is written as Figure 4, but the correct description is Figure 5.

Author Response

We thank the reviewer for the positive opinion of the manuscript.

The authors synthesized Eosin yellow-gold-polypyrrole hybrid nanoparticles (E-G-Ppy) and evaluated their physicochemical properties, skin permeability, and usefulness after the application of the gel formulation to patients. This study is interesting because it demonstrates the efficacy of the combined photodynamic/photothermal effect of polypyrrole-loaded eosin. The paper also accurately compared the results with previous studies. However, the following comments need to be addressed.

-Comment 1:

  1. With Eosin, the zeta potential changed from positive to negative. Drug penetration into the skin may be highly dependent on the charge of the formulation. The skin penetration and clinical results of the present study should be discussed from the zeta of the formulation.

-Reply:

We thank the reviewer for this comment. In fact, zeta potential of suspended nanoparticles (or any colloidal system) is a critical factor that determines the degree of skin penetration. But the colloidal solution, in this work, was not applied in its crude form. It was formulated, like the other applied formulations, in the form of gel. Since the synthesized nanoparticles were embedded in the polymer network (carboxymethylcellulose) forming a gel within which they cannot move freely, zeta potential values in the new dosage forms cannot be estimated and hence, the zeta potential values of the crude colloidal solutions are not applicable or indicative to the gel forms. Therefore, we suggested that the penetration into the skin layers depends mainly on particle size rather than to the zeta potentials of the original colloidal solutions of the formulae.

-Comment 2:

  1. In the description of Figure 3, black stars and red arrows are absent and should be deleted.

-Reply:

Thanks for the comment. They were deleted in the modified manuscript (Lines: 229-230).

-Comment 3:

  1. Figure 4 shows changes in skin properties such as inflammation, but are these results transient? If these changes persist, it is a negative clinical finding. The authors need to explain their findings.

-Reply:

We thank the reviewer for this comment. In fact, the irradiation power used in the trans-dermal permeation animal study was much higher than that used clinically and irradiation was carried out using the whole spectrum of white light to pre-assure the maximum safety. However, post-irradiation inflammation is common in photodynamic and photothermal therapy. These changes are usually temporary and relieved as the treatment is ended. Nevertheless, Hidradenitis suppurativa itself is “a type of chronic suppurative inflammatory reaction of the hair follicles characterized by recurrent dermal abscesses, sinus tracts and scars”1. PDT and PTT are based on singlet oxygen-induced or heat-induced destruction of the diseased apocrine glands hair follicles leading to healing through eradication of the disease causes. PDT and PTT have high selectivity due to the localization of effect at the area of application. Due to its polymeric nature, Ppy enhances the penetration of EY (hydrophilic PS) and protect it from rapid clearance.

-Comment 4:

  1. In addition, HAuCl4 in the gel formulation is expected to remain in the body as gold nanoparticles after application to the patient. The safety of this formulation must also be demonstrated.

-Reply:

We thank the reviewer for this comment. In fact, there is not any HAuCl4 in any of the formulations, gold is present as ultra-nano gold nanoparticles embedded within the either Ppy or E-Ppy matrix. The prepared nanoparticles and nanocomposites (G-Ppy and E-G-Ppy) were prepared, washed many times and assessed, in a previous paper, for their dark cytotoxicity and were found to be non-toxic even at high doses.2 We expected that any side effects, if present, will be at their minimum.  So, the topical use of the nanoparticles didn’t report any side effect after the follow up period monitored by the dermatologist. 

-Comment 5:

  1. The description of the Figure in the Clinical case study is written as Figure 4, but the correct description is Figure 5.

-Reply:

Thanks for the comment. This mistake was corrected. All figures and tables were revised and adjusted.

References

1           Zhang, Y., Yang, Y. & Zou, X. Photodynamic therapy for Hidradenitis Suppurativa/acne inversa: Case report. Photodiagnosis and Photodynamic Therapy 22, 251-252, doi:https://doi.org/10.1016/j.pdpdt.2018.04.014 (2018).

2           El-Kholy, A., Abdel Fadeel, D., Nasr, M., Fadel, M. & El-Sherbiny, I. (Rose Bengal)/(Eosin Yellow)-Gold-Polypyrrole Hybrids: A Design for Dual Photo-Active Nano-System with Ultra-High Loading Capacity. Drug Design, Development and Therapy Volume 15, 5011-5023, doi:10.2147/DDDT.S338922 (2021).

Reviewer 3 Report

The research study” Polypyrrole-loaded Eosin in photo-mediated treatment of hidradenitis suppurativa: In vivo trans-epidermal permeation 3 study and clinical case report” directed to for Hidradenitis suppurativa management that depends on the synergetic photodynamic and photothermal effect of the synthesized Eosin yellow gold polypyrrole hybrid nanoparticles (E-G-Ppy). E-G-Ppy and Gold-Polypyrrole NPs (G-Ppy) 20 were synthesized, characterized and prepared in topical hydrogel. The study is brief and extension of pervious research done by author/s.

The observation and comments are as follows:

1. Line 39-40: Photodynamic therapy (PDT) was studied as a promising new technique for the treatment of HS but the results were controversial [4-6]. Can author add on this…about controversy in one or two sentences.

2. Introduction: Why gold use for nanoparticle? Brief in introduction answering what and why?

3.  Fig 1: x axis legend, must same as in figure or text description for better understating of readers.

4. Fig 2. TEM imaging of the free G-Ppy (fig. 2A) showed shielded clusters with a mean diameter of 50-60 nm for each cluster. Gold nanoparticles image showed diameter range between 3-12 nm..etc. From future it is difficult to prove. Please update high resolution size marked image (Specially Fig 2B).

5. Merge Fig 3 and 4.

6. Line 182: is EY-AuPpy is same as EY-G-Ppy or different. Please stick to nomenclature/abbreviation used.

7. Line 185: improvement of 80% of the lesion was achieved, what that mean by and how it is measured?

8. There are two time figure 4, please correct the number in text as well as captions.

9. 187: If six-month study is there of application, mention frequency and dose given.

10. Discussion part, language improvement suggested.

11. Discussion need to improve with inclusion of data of result, and key findings.

12. Compare you results if any with Fractional CO2 Laser before gel application in permeation improvement.

Author Response

We thank the reviewer for the positive opinion of the manuscript.

-Comment 1:

  1. Line 39-40: Photodynamic therapy (PDT) was studied as a promising new technique for the treatment of HS but the results were controversial [4-6]. Can author add on this…about controversy in one or two sentences.

-Reply:

We thank the reviewer for this comment. This was unintended mistake. This word “controversial’ was deleted with the necessary modifications added. (Lines: 75-76)

-Comment 2:

  1. Introduction: Why gold use for nanoparticle? Brief in introduction answering what and why?

-Reply:

We thank the reviewer for this comment. The necessary information was added. (Lines: 87-90)

-Comment 3:

  1. Fig 1: x axis legend, must same as in figure or text description for better understating of readers.

-Reply:

We thank the reviewer for this comment. The word “intensity” was added to the axis legend to be “absorption intensity”. (The modified fig.1, lines: 195-196)

-Comment 4:

  1. Fig 2. TEM imaging of the free G-Ppy (fig. 2A) showed shielded clusters with a mean diameter of 50-60 nm for each cluster. Gold nanoparticles image showed diameter range between 3-12 nm..etc. From future it is difficult to prove. Please update high resolution size marked image (Specially Fig 2B).

-Reply:

Thanks for the comment. Indeed, this misunderstanding arises from the misleading figure caption which is not accurately defined. This was unintended oversight. Another well-illustrative figure caption was added to replace the old one. Also, the TEM image was updated and adjusted for resolution. (The modified fig.2, lines: 200-204 and lines: 191-193)

-Comment 5:

  1. Merge Fig 3 and 4.

-Reply:

The two figures were merged into fig.3 in the modified manuscript. (The modified fig.3, lines: 226-227)

-Comment 6:

  1. Line 182: is EY-AuPpy is same as EY-G-Ppy or different. Please stick to nomenclature/abbreviation used.

-Reply:

They are the same. This was an unintended mistake. This was corrected in the modified manuscript and all the same-indicative abbreviations along the manuscript were unified to avoid distraction. (Line: 243)

-Comment 7:

  1. Line 185: improvement of 80% of the lesion was achieved, what that mean by and how it is measured?

-Reply:

Thanks for the comment. The criteria of improvement evaluation were thoroughly discussed in the section Materials and methods. (Lines: 182-185)

-Comment 8:

  1. There are two time figure 4, please correct the number in text as well as captions.

-Reply:

This was corrected in the modified manuscript, taking in consideration the merging between fig.3 and fig.4 into one figure (fig.3). (Lines: 226-227)

-Comment 9:

  1. 187: If six-month study is there of application, mention frequency and dose given.

-Reply:

Thanks for the comment. The six-month period was a post-treatment follow up period not a treatment period.

-Comment 10:

  1. Discussion part, language improvement suggested.

-Reply:

Thanks for the comment. The whole paper was revised for language and necessary modifications were done.

-Comment 11:

  1. Discussion need to improve with inclusion of data of result, and key findings.

-Reply:

Thanks for the comment. Discussion was revised, cleared and enriched in the modified manuscript. (Lines: 254-299)

-Comment 12:

  1. Compare you results if any with Fractional CO2 Laser before gel application in permeation improvement.

-Reply:

We thank the reviewer for this comment. In fact, CO2 laser step is just to enhance the penetration of the applied gel. However, applying CO2 laser alone without treatment had insignificant effects. We decided to list this step in the protocol just to guarantee achieving maximum benefit for the patient according to Helsinki declaration. Thus, this step, alone, has insignificant effects.
